# Methodological considerations for linking household and healthcare provider data for estimating effective coverage: a systematic review

Emily D Carter [ID],[1] Hannah H Leslie [ID],[2] Tanya Marchant,[3] Agbessi Amouzou,[1] Melinda K Munos [ID] [1]

[1]International Health, Johns Hopkins University Bloomberg School of Public Health, Baltimore, Maryland, USA
[2]Global Health and Population, Harvard TH Chan School of Public Health, Boston, Massachusetts, USA
[3]Disease Control, London School of Hygiene and Tropical Medicine, London, UK

**Correspondence to**
Dr Emily D Carter;
ecarter@jhu.edu

## ABSTRACT

**Objective** To assess existing knowledge related to methodological considerations for linking population-based surveys and health facility data to generate effective coverage estimates. Effective coverage estimates the proportion of individuals in need of an intervention who receive it with sufficient quality to achieve health benefit.

**Design** Systematic review of available literature.

**Data sources** Medline, Carolina Population Health Center and Demographic and Health Survey publications and handsearch of related or referenced works of all articles included in full text review. The search included publications from 1 January 2000 to 29 March 2021.

**Eligibility criteria** Publications explicitly evaluating (1) the suitability of data, (2) the implications of the design of existing data sources and (3) the impact of choice of method for combining datasets to obtain linked coverage estimates.

**Results** Of 3805 papers reviewed, 70 publications addressed relevant issues. Limited data suggest household surveys can be used to identify sources of care, but their validity in estimating intervention need was variable. Methods for collecting provider data and constructing quality indices were diverse and presented limitations. There was little empirical data supporting an association between structural, process and outcome quality. Few studies addressed the influence of the design of common data sources on linking analyses, including imprecise household geographical information system data, provider sampling design and estimate stability. The most consistent evidence suggested under certain conditions, combining data based on geographical proximity or administrative catchment (ecological linking) produced similar estimates to linking based on the specific provider utilised (exact match linking).

**Conclusions** Linking household and healthcare provider data can leverage existing data sources to generate more informative estimates of intervention coverage and care. However, existing evidence on methods for linking data for effective coverage estimation are variable and numerous methodological questions remain. There is need for additional research to develop evidence-based, standardised best practices for these analyses.

## STRENGTHS AND LIMITATIONS OF THIS STUDY

⇒ We systematically reviewed a wide range of methodological issues pertaining to linking population-based and health provider data for effective coverage estimation.
⇒ The review was limited by the diversity of terminology and fields related to the linking methodology.
⇒ Multiple search strategies were used to minimise the likelihood of overlooking relevant publications.
⇒ Results of the review are summarised and related to actionable items and needs for future research.

## BACKGROUND

There is growing demand for tracking progress towards the sustainable development goals through effective coverage estimates.[1 2] Effective coverage measures assess not only the proportion of individuals in need of an intervention who receive it, but also the content and quality of services received with an aim to estimate the proportion of individuals receiving the health benefit of an intervention.[2] Numerous publications have estimated effective coverage[3] using a range of methods and measures to define intervention need, receipt and quality.

Linking household and health provider data is a promising means of generating effective coverage estimates that provide population-based estimates and incorporate data on service quality from health facilities. Data from household surveys can provide a population-based estimate of intervention need and care-seeking for services, such as the proportion of women with a recent live birth who delivered in a health facility. However, a number of maternal, newborn and child health interventions[4 5] cannot be accurately measured through household surveys due to reporting errors and biases by respondents (eg, the proportion of women who received a

uterotonic during delivery). Health provider assessments yield information on provider quality, including available infrastructure, commodities, equipment, human resources and potentially provision of care. Provider data do not capture need for care in the population, care-seeking behaviour or the experience of individuals who do not access the formal health system. Linking these two data sources can provide a more complete picture of population access to and coverage of high-quality health services, for example, the proportion of women who delivered at a health facility with sufficient structural resources and competence to provide appropriate labour and delivery care.

There are many approaches for combining household and provider datasets.[6] The results depend on the choice of data and of methods for combining datasets. However, very limited guidance exists to guide decision making. We conducted a systematic review to understand the current evidence base for effective coverage linking methods and identify needs for further research.

## METHODS

We searched for papers addressing methods or assumptions regarding: (1) the suitability of household and provider (defining health providers as healthcare outlets such as health facilities, pharmacies, and community-based health workers) data used in linking analyses, (2) the implications of the design of existing household (Demographic and Health Survey (DHS) and Multiple Indicator Cluster Survey (MICS)) and provider (Service Provision Assessment (SPA) and Service Availability and Readiness Assessment (SARA)) data sources commonly used in linking analyses and (3) the impact of choice of method for combining datasets to obtain linked coverage estimates.

Our primary search was conducted in Medline. The search was limited to papers published between 1 January 2000 and 29 March 2021 that included terms related to (1) effective coverage, benchmarking, system dynamics or universal health coverage (UHC) metrics, or (2) structural, process and/or health outcome quality, (3) linking analyses using terms adapted from Do et al,[6] (4) validity of self-report health indicators and (5) spatial methods for measuring utilisation or distance to care. A full list of Medline search terms is presented in online supplemental file 1. The search was conducted using English-language terms; however, publications in English, Spanish and French were reviewed if captured in the search. Additionally, we conducted searches using these criteria in Population Health Metrics (which was not fully indexed in Medline at the time of our search), the Carolina Population Health Center and DHS publications. In a second step, we handsearched the references of a systematic review by Do et al on linking household and facility data to estimate coverage of reproductive, maternal, newborn and child health (RMNCH) services,[6] and a review by Amouzou et al of effective coverage analyses.[3] Both the

Do and Amouzou reviews summarised publications that linked data or estimated effective coverage; however, they did not systematically address methodological concerns or relevant results for guiding application of these methods. We also handsearched the references, citing works and journal—or database interface-generated related publications of all articles that passed the title and abstract review.

Publications were reviewed for relevant analyses or commentary related to linking methodologies. Articles were included if they explicitly evaluated or compared assumptions used in linking approaches for at least one of the areas defined above. The review focused on low-income and middle-income countries (LMICs) and data sources common in these settings, however, publications from high-income settings were retained if the relevant evidence could translate to LMICs (eg, use of centroid global positioning system (GPS) location in estimates of distance, validity of provider quality measures). No formal quality assessment was conducted due to the diversity of study designs and research objectives of the papers relevant to the review. Title and abstract review were conducted simultaneously by the first author (EC). Data extraction included the title, author, year of publication, country or countries included in analysis, data source and specific analyses or findings relevant to linking loosely categorised by topic areas. Topical area groupings emerged from the review and were used to structure the findings.

### Patient and public involvement

As a systematic review, neither patients nor the public were involved in the design, conduct, reporting, or dissemination plans of our research.

### RESULTS

The Medline search produced 3669 publications, along with 79 from the Carolina Population Center, 4 from Population Health Metrics, 12 DHS publications, 35 papers included in the review by Amouzou et al and 49 papers included in the review by Do et al meeting the publication date restrictions. After removing duplicates, 3805 publications were included in the title and abstract review and 236 were included in the full text review. Of those papers included in the full text review, 56 publications addressed a methodological concern related to linking household and provider data and were included in the final review. Fourteen additional publications were identified through the snowball review of references and related works (figure 1). In total 70 publications addressed one or more methodological concern, including the suitability of household (n=13) and provider data (n=39) for use in linking analyses, concerns related to the design of existing household (n=6) and provider (n=4) data sources and methods for combining household and facility data (n=14). A list of publications included in the review and a summary of their contributions to the review are provided in table 1.

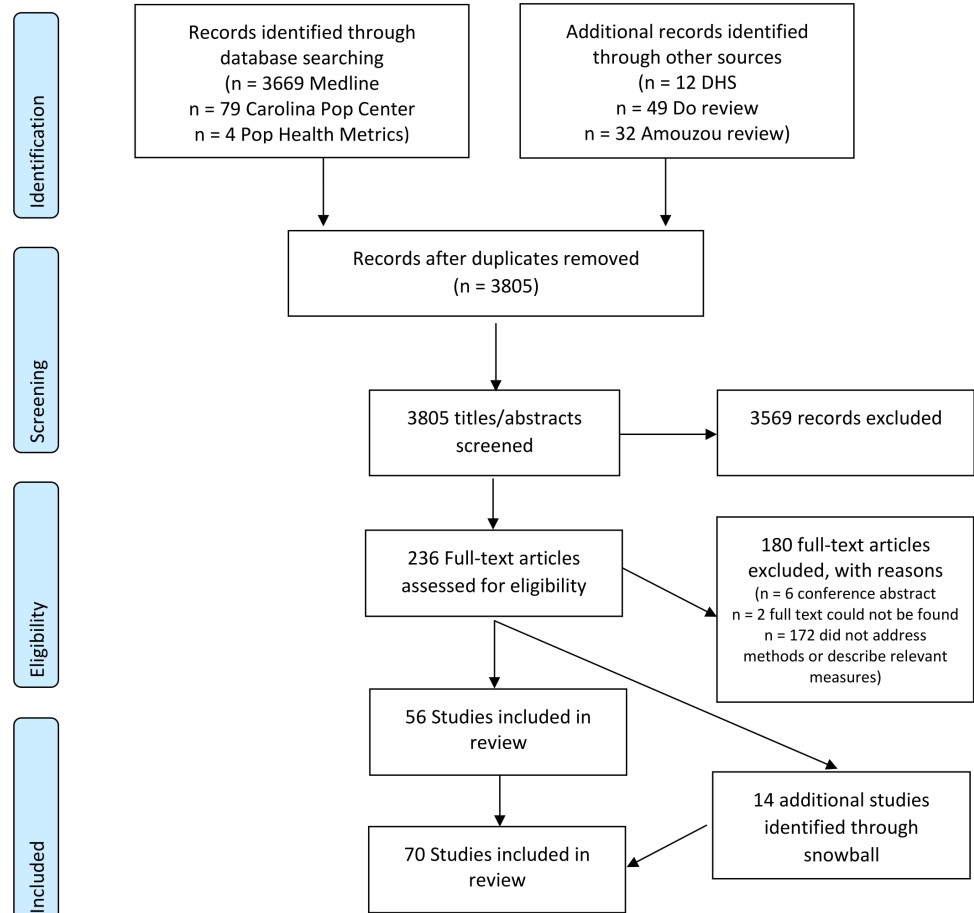

**Figure 1** PRISMA flow diagram. DHS, Demographic and Health Survey; PRISMA, Preferred Reporting Items for Systematic Reviews and Meta-Analyses.

## Suitability of household and provider data for linking analyses

### Suitability of household data needed for linked estimates

In effective coverage linking analyses, household surveys can be used to estimate the proportion of the population in need of healthcare, as well as care-seeking behaviour. Household surveys must produce valid estimates of these parameters and provide care-seeking data that can be linked to provider assessments. This review identified papers discussing issues in defining intervention need (n=8) and care-seeking (n=5) that should guide selection of indicators for linking.

### Intervention need

Estimation of intervention need may require solely population demographics such as age (eg, for prevention and health promotion interventions) or may require defining specific illnesses or conditions. The latter is more subject to reporting bias.[7] Multiple studies have shown poor association or biases between maternally reported symptoms and clinical pneumonia,[8 9] malaria[10] and diarrhoea[11] in children under 5. A handful of studies (n=3) showed maternal report of both maternal and newborn birth complications is variable.[12–14] A simulation by Shengelia et al demonstrated the effect of the divergence of true from perceived intervention need on effective coverage estimates. The authors propose estimating the posterior

probability of disease based on responses to symptomatic questions using a Bayesian model to measure disease presence on a probabilistic scale.[7] However, there has been no work on how to integrate these adjusted estimates into effective coverage estimates.

### Care-seeking behaviour

Four studies addressed the accuracy of respondent report of seeking care. Mothers in Zambia and Mozambique were able to accurately report on the type of health provider where they sought sick child[15] and delivery care,[16] respectively. However, studies in two countries suggested women cannot report on the type of health worker who attended to them during labour and delivery and immediate postnatal care.[17 18] Wang et al note that provider categories are not standardised between population surveys and health system assessments, with population surveys often including vague or overly broad categories that do not directly match SPA/SARA categories and require harmonisation.[19]

### Suitability of healthcare quality data needed for linked estimates

Provider assessments present data on service content and quality for effective coverage linking analyses. However, the measurement, construction and interpretation of provider quality measures are highly variable and may significantly alter

**Table 1** Summary of publications included in the review and contribution to the literature

| Author | Year | Country | Key method contribution |
|---|---|---|---|
| **Suitability of houseold and provider data for linking analyses**<br>▶ **Household data needed for linked estimates** | | | |
| Ayede et al[8] | 2018 | Nigeria | Accuracy of maternal report of pneumonia symptoms measured through household survey. |
| Blanc et al[17] | 2016 | Mexico | Accuracy of maternal report of delivery/immediate PNC attendant measured through household survey. |
| Blanc et al[18] | 2016 | Kenya | Accuracy of maternal report of delivery/immediate PNC attendant measured through household survey. |
| Carter et al[15] | 2018 | Zambia | Accuracy of maternal report of care-seeking for child illness measured through household survey. |
| Chang et al[14] | 2018 | Nepal | Accuracy of maternal report of birth weight and preterm birth measured through household survey. |
| D'Acremont et al[10] | 2010 | SSA | Reduced proportion of fever cases that are malaria. |
| Hazir et al[9] | 2013 | Pakistan and Bangladesh | Accuracy of maternal report of pneumonia symptoms measured through household survey. |
| Keenan et al[12] | 2017 | USA | Accuracy of maternal recall of birth complications. |
| Shengelia et al[7] | 2005 | – | Effect of true versus perceived intervention need on effective coverage estimation. |
| Stanton et al[16] | 2013 | Mozambique | Accuracy of maternal report of place of delivery measured through household surveys. |
| Fischer Walker et al[11] | 2013 | – | Issues with measurement of child diarrhoea through household surveys. |
| Wang et al[19] | 2018 | Multiple Regions | Issues with provider categories and alignment between DHS and SPA surveys. |
| Zimmerman et al[13] | 2019 | Ethiopia | Reliability of maternal report of maternal and newborn birth complications. |
| **Suitability of household and provider data for linking analyses**<br>▶ **Provider data needed for linked estimates** | | | |
| Akachi and Kruk[57] | 2017 | – | ▶ Need for global benchmarks for quality.<br>▶ Lack on data linking quality with health outcomes. |
| Carter et al[33] | 2018 | Zambia | Quality score for child health effective coverage. |
| Chou et al[39] | 2019 | Multiple Regions | Quality score for maternal and neonatal health effective coverage. |
| Davis et al[32] | 2006 | High-income countries | Agreement between provider self-assessment and observed quality. |
| Diamond-Smith et al[24] | 2016 | Kenya and Namibia | Association between maternal perception of care and measured structural and process quality. |
| Fisseha et al[51] | 2017 | Ethiopia | Internal consistency of structural and process quality indicator. |
| Gabrysch et al[45] | 2011 | Zambia | Quality score for labour and delivery effective coverage. |
| Getachew et al[25] | 2020 | Ethiopia | Association between caregiver perception of care and measured structural and process quality. |
| Hoogenboom et al[27] | 2015 | Thai-Myanmar Border | Agreement between facility records and observed care. |
| Hrisos et al[30] | 2009 | High-income countries | Systematic review of agreement between observed quality of care and provider self-report, patient-report, and/or chart review. |
| Jackson et al[53] | 2015 | Tanzania | PCA to reduce quality index. |
| Joseph et al[42] | 2020 | Malawi | ▶ Quality score for ANC nutrition effective coverage.<br>▶ Association between quality-adjusted coverage and LBW. |
| Kanyangarara et al[37] | 2017 | SSA | Quality score for ANC effective coverage. |

Continued

**Table 1** Continued

| Author | Year | Country | Key method contribution |
|---|---|---|---|
| Kruk et al[55] | 2017 | SSA | Association between structural and process quality. |
| Larson et al[26] | 2014 | Tanzania | ▲ Association between maternal perception of care and service availability and respect.<br>▲ Vignettes for measuring quality. |
| Leegwater et al[59] | 2015 | – | Association between UHC index and infant mortality and life expectancy at national level. |
| Leslie et al[58] | 2016 | Malawi | Association between quality of delivery care and neonatal mortality. |
| Leslie et al[41] | 2017 | SSA | ▲ Quality score for ANC, labour and delivery, sick child, and family planning effective coverage.<br>▲ Association between structural and process quality. |
| Leslie et al[54] | 2018 | Multiple Regions | Performance of approaches for generating service readiness indices. |
| Leslie et al[46] | 2019 | Mexico | Quality score for ANC, labour and delivery, newborn, sick child, chronic conditions, and cancer treatment effective coverage. |
| Lozano et al[49] | 2006 | Mexico | UHC index using weighted vs simple average of indicators |
| Mallick et al[52] | 2017 | Haiti, Malawi and Tanzania | Comparison of measures of family planning quality. |
| Marchant et al[35] | 2015 | Ethiopia, Nigeria and India | Measurement of quality using "last delivery module". |
| Mboya et al[36] | 2016 | Tanzania | mHealth tool to measure quality. |
| MCSP[23] | 2018 | Multiple Regions | Availability and quality of data captured through HMIS. |
| Moucheraud and McBride[48] | 2020 | SSA and Haiti | Systematic review of quality measures derived from SPA data. |
| Munos et al[40] | 2018 | Cote D'Ivoire | Quality score for ANC, labour and delivery, PNC, and child health effective coverage. |
| Nesbitt et al[28] | 2013 | Ghana | Quality score for labour and delivery and PNC effective coverage. |
| Nguhiu et al[38] | 2017 | Kenya | Quality score for ANC, labour and delivery, sick child, and family planning effective coverage. |
| Nickerson et al[21] | 2015 | Multiple Regions | Comparison of data collected through health facility assessments. |
| Osen et al[29] | 2011 | Ghana | Agreement between provider reported and observed surgical service quality. |
| Peabody et al[31] | 2000 | US | Agreement between vignettes, chart abstraction, and simulated client measures. |
| Serván-Mori et al[47] | 2019 | Mexico | Quality score for labour and delivery and newborn care effective coverage. |
| Sheffel et al[22] | 2018 | Multiple regions | Summary of quality data collected through SPA and SPA. |
| Sheffel[50] | 2018 | Haiti, Malawi, Tanzania | Association between structural and process quality. |
| Willey et al[44] | 2018 | Uganda | Quality score for labour and delivery and newborn care effective coverage. |
| Wilunda et al[34] | 2015 | Uganda | Quality score for maternal and neonatal care effective coverage. |
| Zurovac et al[56] | 2015 | Vanuatu | Poor association between structural quality and clinical care in fever management. |
| **Implications of design of existing household and provider data sources commonly used in linking analyses** | | | |
| ▲ **Household data** | | | |
| Bliss et al[62] | 2012 | USA | Comparison of distance using centroid vs true location. |

Continued

**Table 1** Continued

| Author | Year | Country | Key method contribution |
|---|---|---|---|
| Healy and Gilliland[64] | 2012 | Canada and UK | Comparison of distance using centroids of varying areal groupings. |
| Jones et al[63] | 2010 | USA | Comparison of distance using zip-code centroid versus true household location. |
| Nesbitt et al[65] | 2014 | Ghana | Comparison of straight-line distance, network distance, raster and network-based travel time distance measures using village versus compound centroid. |
| Perez-Heydrch et al[60] | 2013 | – | Effect of DHS cluster displacement on distance measures. |
| Skiles et al[66] | 2013 | Rwanda | Effect of DHS cluster displacement on estimates of service environment. |
| **Implications of designs of existing household and provider data sources commonly used in linking analyses** | | | |
| ▶ **Provider data** | | | |
| Carter et al[33] | 2018 | Zambia | Effect of excluding non-facility providers from sampling frame on effective coverage estimates. |
| Munos et al[40] | 2018 | Cote d'Ivoire | Effect of excluding non-facility providers from sampling frame on effective coverage estimates. |
| Skiles et al[66] | 2013 | Rwanda | Effect of facility sampling on estimates of service environment. |
| Turner et al[67] | 2001 | – | ▶ Limitations of SPA sampling design. <br> ▶ Approach for joint sampling of households and facilities for linking analyses. |
| **Implications of designs of existing household and provider data sources commonly used in linking analyses** | | | |
| ▶ **Survey timing** | | | |
| Baker et al[68] | 2005 | Uganda and Tanzania | Stability of facility diagnostic capacity over time. |
| Marchant et al[69] | 2008 | Tanzania | Stability of IPTp stocks. |
| Wang et al[71] | 2011 | Multiple Regions | Stability of maternal healthcare-seeking behaviours measured through household survey over time. |
| Willey et al[44] | 2018 | Uganda | Stability of facility infrastructure indicators for labour, delivery, and newborn care. |
| Winter et al[70] | 2015 | Multiple Regions | Stability of care-seeking for child illness behaviours measured through household survey over time. |
| **Impact of choice of method for combining household and provider data** | | | |
| ▶ **Comparison of exact match and ecological linking methods for estimating effective coverage** | | | |
| Carter et al[33] | 2018 | Zambia | Comparison of exact match and ecological linking methods in estimating effective coverage in sick child care. |
| Munos et al[40] | 2018 | Cote d'Ivoire | Comparison of exact match and ecological linking methods in estimating effective coverage in ANC, labour and delivery, PNC and sick child care. |
| Willey et al[44] | 2017 | Uganda | Comparison of exact match and ecological linking methods in estimating effective coverage in ANC, labour and delivery, PNC and sick child care. |
| **Impact of choice of method for combining household and provider data** | | | |
| ▶ **Performance of measures of geographical proximity for ecological linking** | | | |
| Carter et al[33] | 2018 | Zambia | Comparison of true-source of care for child illness to straight-line and road distance measures. |
| Delamater et al[74] | 2019 | US | Comparison of FCA, simple distance, and Huff distance measure against true utilisation patterns. |
| Gething et al[76] | 2004 | Kenya | Comparison of Theissen boundaries and true utilisation patterns. |

Continued

**Table 1** Continued

| Author | Year | Country | Key method contribution |
|---|---|---|---|
| Munos et al[40] | 2018 | Cote d'Ivoire | Comparison of true-source of care for ANC, labour and delivery, PNC and child illness to straight-line and road distance measures. |
| Noor et al[72] | 2006 | Kenya | Comparison of true-source of care for child fever to closest by Euclidian and road distance. |
| Tanser et al[77] | 2001 | South Africa | Comparison of Theissen boundaries and true utilisation patterns. |
| Tanser et al[73] | 2006 | South Africa | Comparison of typical source of care to closest by travel time. |
| Tsoka and Le Sueur[75] | 2004 | South Africa | Comparison of Theissen boundaries and true utilisation patterns. |
| **Impact of choice of method for combining household and provider data** | | | |
| ▲ Statistical challenges | | | |
| Sauer et al[79] | 2020 | – | Comparison of exact, parametric bootstrap and delta method for estimating effective coverage variance. |
| Wang et al[19] | 2018 | Multiple Regions | Use of Delta method for estimating effective coverage variance. |
| Willey et al[44] | 2018 | Uganda | Use of Delta method for estimating effective coverage variance. |

ANC, antenatal care; DHS, Demographic and Health Survey; FCA, floating catchment area; HMIS, Health Management Information Systems; IPTp, intermittent preventive treatment of malaria in pregnancy; LBW, low birthweigh; MCSP, Maternal and Child Survival Program; PCA, principal component analysis; PNC, postnatal care; SPA, Service Provision Assessment; SSA, sub-Saharan Africa; UHC, universal health coverage.

effective coverage estimates. This paper does not present an exhaustive review of healthcare quality measures or the association between levels of quality. A comprehensive summary of quality of care concepts and measurement approaches, along with their relative strengths and limitations, was presented by Hanefeld et al.[20] Publications of particular relevance to linking analyses are noted here, with an emphasis on national provider survey data as the most common source of provider data for linking analyses.

### Methods used in assessing provider quality

A review by Nickerson et al found significant variability in the data collected and methods used in health facility assessment tools in LMICs.[21] While SPA and SARA data are the most widely used sources of data on health service delivery in LMICs, one paper noted that these surveys focused primarily on structural quality with less data on provision and experience of care.[22] The lack of process quality data is in part related to the reliance on direct observation of clinical care—a time-intensive and resource-intensive method—to collect these data. None of the studies included in the review used Health Management Information Systems (HMIS) data to generate linked coverage estimates. A desk review by the Maternal and Child Survival Program (MCSP) found that data collected through HMIS was variable across countries, data recorded within registers often was not transmitted through the system, and only a limited number of indicators collected were related to the provision of health services.[23]

Nine publications assessed alternatives to direct observation of clinical care for collecting process quality data. Two studies found no association between process quality and maternal perceptions of the quality of care received[24 25] while one study found perceived quality was associated with the number of services received but not structural quality.[26] Agreement between observed care and health records or provider report was also variable.[27–29] A review by Hrisos et al found few studies to support use of patient report, provider self-report or record review as proxy measures of clinical care quality.[30] In the USA, vignettes performed better than chart abstraction for estimating quality.[31] Another review found providers were unable to accurately assess their own performance, with the worst accuracy among the least skilled providers.[32] Five other publications used alternative methods for measuring process quality, including use of vignettes,[26 33] register review,[26 34] most recent delivery interview[35] and an mHealth tool,[36] but did not assess their performance against other measurement methods.

### Content of provider quality indices

Most linking papers estimating effective coverage included in this review (n=15) characterised provider quality using structural measures of quality, with or without measures of process quality. Various approaches were used to select items for inclusion in these measures. Measures of structural and process quality were derived from either national or international guidance on minimum service availability and required commodities, equipment,

infrastructure, training or actions. Measures used by effective coverage analyses included SPA or SARA structural indicators[33 37–39] and/or clinical observations,[38 40–43] emergency obstetric and newborn care functions,[28 34 44 45] provider recall of actions during their last delivery,[35 44] and measured health outcomes.[46 47]

### Construction of provider quality indices

In addition to the range of variables used in provider measures, there was no consensus on the approach to use to select and combine variables to generate quality indices. The reviewed publications used a variety of approaches to construct indices including weighted indices,[41] simple averages across all indicators or domains,[33 39 40 42–44] and categorisation using set thresholds or relative categories.[26 37 45] A review of quality measurement using SPA data found that studies frequently did not apply a theoretical framework when selecting indicators for quality measures, and that there was high variability in the indicators included in quality scores.[48] In our review, seven publications presented data on the performance of different measurement modes and summary approaches. Two studies found the method of selecting and combining quality indicators had little effect on overall effective coverage estimates.[49 50] However, two other studies found inconsistency in the rankings of health facilities when using different index methods.[51 52] Two studies using principal component analysis (PCA) to create SPA health service indices found the reduced indices explained only a limited amount of the variance across indicators.[52 53] An analysis of SPA data in ten countries found indices empirically derived through machine learning captured a large proportion of the service readiness data in the full SPA index, however, the selected set of indicators varied across countries, and an index generated through expert review captured very little of the data from the full index.[54] Two studies found that few facilities could meet all requirements when applying a threshold, limiting the utility of the approach.[45 51]

### Performance of provider measures

Despite the common usage of SPA and SARA data-derived structural and process quality measures, the review found limited data explicitly assessing the association of these measures with each other and health outcomes (n=7). Three studies, two incorporating data from multiple countries, found little association between structural quality and process quality.[41 55 56] However, an analysis of SPA data from three countries found a small but significant association between antenatal care (ANC) facility structural and process quality and suggests structural quality can limit provider performance when basic infrastructure and commodities are unavailable.[50] Akachi and Kruk emphasised the limited number of studies showing process quality associated with health outcomes.[57] Two studies in Malawi found a small association between an obstetric quality index and decreased neonatal mortality[58] and an association between quality-adjusted

ANC nutrition intervention coverage and decreased low birth weight prevalence.[42] Another found a national UHC 'heath service coverage' index correlated strongly with infant mortality rate and life expectancy.[59]

### Implications of design of existing household and provider data sources commonly used in linking analyses

#### Issues related to household and cluster location data

The way in which common household surveys, particularly the DHS and MICS, collect and process location data may also impact the validity of some linked estimates. In many household datasets used for linked analyses, the precise location of individual households is often unknown. The DHS collects central point locations for clusters, rather than household locations, and displaces these points in publicly released datasets.[60] MICS often does not collect or make geographical information system (GIS) data available.[61] Imprecision around household location may influence the accuracy of estimates generated by linking household and provider data based on geographical proximity.

### Data on household location

The effect of using cluster central point locations rather than individual household locations in linking analyses was not addressed by any publication identified in this review. However, four studies looked at the effect of using centroids of varying areal units versus household locations in distance analyses. Two studies found using US census tract[62] and zip-code[63] centroid locations produced little difference in measures of facility access compared with household location. A third study showed use of areal unit centroids resulted in misclassification of household access to health-related facilities, especially in less densely populated rural areas.[64] However, in rural Ghana, measures calculated from village centroids identified the same closest facility as measures from compound locations for over 85% of births.[65]

### Cluster displacement

Displacement of cluster central points might induce additional error in analyses based on geographical proximity. A DHS analytical report found that ignoring DHS displacement in analyses that used distance to a resource as a covariate resulted in increased bias and mean squared error. However, this will not affect linking by administrative unit because DHS has restricted displacement to within the representative sample administrative unit since 2009.[60] A simulation analysis in Rwanda reported DHS cluster displacement produced less misclassification in level of access and relative service quality than healthcare provider sampling.[66]

#### Issues related to provider sampling

Typical sampling designs for healthcare provider data also present issues for linking analyses. Both SPAs and SARAs are sampled independently of household surveys, thus, there may be no sampled facilities near household survey clusters.[67] SPA and SARA surveys typically collect data on a sample,

rather than census, of public, private and non-governmental organization (NGO) health facilities and exclude non-facility providers, such as pharmacies or community health workers (CHWs). In most settings, facilities are sampled and analysed to be representative of all facilities within a managing authority, level and/or geographical area, and the results of the provider assessment are not intended to represent the population using health services.[67] For provider assessments conducting direct observations of clinical care, the number and type of interactions observed within each health facility is dependent on patient volume and chance.

### Provider sampling frame

Two papers assessed the impact of excluding non-facility providers on linked effective coverage estimates. In Zambia and Cote d'Ivoire, CHWs offered a level of care for sick children similar to first-level public facilities. Excluding these providers reduced estimates of effective coverage in Zambia where CHWs were a significant source of skilled care in rural areas,[33] but had little effect in Cote d'Ivoire where they were an insignificant source of care.[40] In both studies, exclusion of pharmacies did not alter effective coverage estimates as they were an uncommon source of care, though they offered moderate structural quality.[33 40]

### Provider sampling design

Two publications addressed the impact of facility survey sampling designs. At the facility level, Skiles *et al*'s analysis demonstrated that sampling facilities, rather than using a census, led to an underestimation of the adequacy of the health service environment and substantial misclassification error in relative service environment for individual clusters.[66] No studies addressed the suitability of SPA or SARA facility sampling strategies for generating stable quality estimates for use in linking analyses at a level below administrative unit used for the sampling approach.

A Measure Evaluation manual emphasised that data on provision of services (collected through observation of client–staff interactions), experience of care (collected through client exit interviews) and staff characteristics (collected through health worker interviews) are sampled independently and collected among health workers and care interactions available on the day of the survey. These data are a subsample of the overall survey and representative at the level the survey is sampled to be representative—not at the facility level.[67] This paper proposed multiple linked sampling approaches to capture geographically concordant household and provider data for linked analyses. While multiple studies included in this review used a census or sample of providers derived from a household sample, none implemented this approach at a national scale.

### Issues related to timing of surveys used in linked coverage estimates

Both care-seeking behaviour and provider quality are likely to vary over time, and both household and provider surveys are conducted infrequently in LMICs (~3–5 years).

Linked coverage estimates for RMNCH may cover a long time frame as the reference period for care-seeking in household surveys varies from 2 weeks (sick child care) to 2–5 years (peripartum care). Population movement and quality improvement efforts at facilities further complicate associations with increasing time lags. The implications of linking household and provider indicators of different temporal periods is unclear.

### Stability of provider indicators

No paper in this review specifically addressed the effect of provider indicator stability on linked effective coverage estimates. However, three linking papers presented data on the stability of some health facility indicators over time. Expanded Quality Management Using Information Power (EQUIP) studies in Uganda and Tanzania found moderate variability in the availability of some maternal and newborn health commodities and services over a period of 2–3 years.[44 68 69]

### Stability of household indicators

Care-seeking behaviour, including overall rates of care and utilisation of different sources of care, may also change over time. Analysis of care-seeking for child illness[70] and maternal healthcare[71] in multiple LMICs over time showed high inconsistency in trends across countries. However, no identified studies addressed the consequences of this temporal variability within the context of linking analyses.

## Impact of choice of method for combining household and provider data

The approach for combining household and provider data can potentially have a significant impact on linked coverage estimates. Methods used to link data, including exact match and various types of ecological linking, are defined in table 2. Exact match linking assigns provider information to individuals in the target population based on their specific source of care. This approach, while potentially subject to the reporting biases described previously, is considered the most precise approach for combining the two data sets in the absence of individual patient health records.[6] Without data on specific source of care, ecological linking approaches are designed to approximate care-seeking behaviour or model healthcare access by linking the target population to sources of care based on geographical proximity or administrative catchment area, making assumptions about service access and use.

### Comparison of exact match and ecological linking methods for estimating effective coverage

Three publications explicitly compared effective coverage estimates generated using exact match and ecological linking methods (table 3).[33 40 44] Estimates generated using the exact match linking approach were considered the gold-standard measure of effective coverage. All three publications found exact match linked effective coverage estimates were similar to straight-line,[33 40] travel

**Table 2** Table of linking approaches

| Approach | Method |
|---|---|
| Exact match | Link to specific reported source of care. |
| Ecological | Link to one or more providers based on geographical proximity or administrative association. |
| Geographical proximity | |
| Straight-line/ Euclidean distance | Closest by absolute (crow-flies) distance. |
| Manhattan distance | Closest by sum of horizontal and vertical distance between points on a grid (blockwise). |
| Minokowski distance | Closest by weighted average of Euclidean and Manhattan distance. |
| Road distance | Closest by distance along a road (line and joint) network. |
| Raster-based travel time | Closest by travel time between points on a continous grid surface with variable transit speed coefficients in each cell. |
| Network-based travel time | Closest by travel time along a road network accounting for variable speed and road conditions. |
| Buffer | All providers within a defined radius from household. |
| Theissen polygon | Define catchment boundaries based on optimal distance between known providers. |
| Kernel density estimation | Define relative draw of providers over geographical area weighted by a density variable. |
| Interpolated surface | Define continuous surface of provider access or quality by smoothing between provider point data. |
| Floating catchment area | Define catchments for known providers allowing for cross-border use (catchment overlap) and distance decay. |
| Administrative | All providers within administrative unit boundaries. |

time,[33 40] 5 km buffer,[33] 10 km buffer[40] and administrative unit[33 40 44] geolinked estimates for antenatal,[40] labour and delivery,[40 44] postnatal[40] and sick child[33 40] care when linking was restricted by the reported provider category (eg, hospital, health centre, CHW). Distance-restricted linking approaches, such as linking to providers within a 5 km radius, produced inaccurate results if unlinked events were treated as no care.[33] Restriction of geographical linking to only providers within the reported category of care and/or weighting by providers' relative patient volume improved agreement between the exact match and ecological linking estimates.[40 44] All three studies also

used provider data obtained from a census of health facilities, and therefore, the findings may not be applicable when household data are linked to a sample survey of health facilities.

### Performance of measures of geographical proximity for ecological linking

Eight studies assessed the performance of geographical measures for assigning households or individuals to their reported source of healthcare. Four studies in sub-Saharan Africa compared the predicted source of care based on geographical proximity against the true source of care. They found straight-line and road distance performed similarly,[72] high performance of shortest travel time method[73] and better performance of straight-line distance compared with road distance.[33 40] In the USA, a more sophisticated approach (two-stage and three-stage floating catchment area) performed better than alternatives methods in assigning households to their source of care.[74] Three studies in sub-Saharan Africa evaluated use of Theissen boundaries, a method of defining catchment boundaries based on the optimal distance between known providers, in assigning households to the catchment of facilities they used. The studies found high performance in some settings,[75] but poorer performance related to the use of higher-order facilities[76] and influence of public transportation routes.[77]

### Statistical challenges

Most linking analyses that have generated effective coverage estimates by assigning individuals the quality score of the reported or linked source of care have derived estimates of uncertainty based on household sampling error and ignored any sampling error around provider data. However, two analyses used the Delta method[78] for estimating the variance of effective coverage estimates generated by multiplying service use and readiness.[19 44] A simulation study compared three variance estimation methods for linked effective coverage measures (household sampling error alone, parametric bootstrapping and the delta method), and found that all three performed similarly for large samples. However, the delta method produced more valid confidence bounds with smaller samples or when the effective coverage estimate approached either 0 or 100%.[79]

### DISCUSSION

This review found a variable number of publications that addressed the diverse methodological issues related to linking household and provider datasets. A summary of key findings and needs for further research is presented in table 4 and discussed below.

### Suitability of household and provider data for linking analyses

We identified a number of papers that critically assessed household and provider data needed for linking analyses. The limited existing data on respondent-reported care-seeking suggest respondents can identify sources of care if

**Table 3** Exact versus ecological linking estimates for select indicators across studies

| | Willey labour and delivery structural QoC | | Carter child health urban structural QoC | | Carter child health rural structural QoC | | Munos labour and delivery structural QoC | | Munos labour and delivery process QoC | | Munos child health structural QoC | | Munos child health process QoC | |
| --- | --- | --- | --- | --- | --- | --- | --- | --- | --- | --- | --- | --- | --- | --- |
| | % (95% CI) | Relative % Diff | % (95% CI) | Relative % Diff | % (95% CI) | Relative % Diff | % (95% CI) | Relative % Diff | % (95% CI) | Relative % Diff | % (95% CI) | Relative % Diff | % (95% CI) | Relative % Diff |
| **Exact match** | 9.86* (3.2 to 16.5) | REF | 49 (43.6 to 54.5) | REF | 60.3 (55.6 to 65.1) | REF | 37.2 (30.5 to 43.9) | REF | 40.1 (32.9 to 47.3) | REF | 22.9 (18.2 to 27.5) | REF | 16.8 (12.8 to 20.8) | REF |
| **Ecological—geographical** | | | | | | | | | | | | | | |
| Absolute distance | | | | | | | 36.5 (29.5 to 43.5) | -1.9% | 39.8 (32.2 to 47.5) | -0.7% | 18.2* (12.3 to 24.1) | -20.5% | 14.3 (9.1 to 19.6) | -14.9% |
| Absolute distance and provider category* | | | 49.1 (43.7 to 54.6) | 0.2 | 61.1 (56.3 to 65.9) | 1.3% | 37 (30.0 to 44.0) | -0.5% | 39.6 (32.1 to 47.1) | -1.2% | 20.8* (16.1 to 25.4) | -9.2% | 16.5 (12.2 to 20.7) | -1.8% |
| Road distance | | | | | | | 36.8 (30.0 to 44.0) | -1.1% | 40.4 (32.6 to 48.1) | 0.7% | 16* (10.9 to 21.1) | -30.1% | 13.8 (8.5 to 19.1) | -17.9% |
| Road distance & provider category* | | | 48.7 (43.2 to 54.1) | -0.6% | 58.8 (54.1 to 63.5) | -2.5% | 37.5 (30.4 to 44.6) | 0.8% | 40.2 (32.5 to 47.9) | 0.2% | 20.2* (16.0 to 24.4) | -11.8% | 16.5 (12.3 to 21.8) | -1.8% |
| Radius 5km and provider category* | | | 49.2 (43.7 to 54.7) | 0.4% | 59.4 (54.8 to 64.1) | -1.5% | | | | | | | | |
| Radius 10km—unweighted† | | | | | | | 35.3* (29.3 to 41.4) | -5.1% | 39.1 (32.0 to 46.2) | -2.5% | 18.8* (14.9 to 22.7) | -17.9% | 15.7 (12.4 to 19.1) | -6.5% |
| Radius 10km—weighted‡ | | | | | | | 37.5 (30.6 to 44.4) | 0.8% | 39.8 (32.5 to 47.1) | -0.7% | 19.1* (15.1 to 23.0) | -16.6% | 15.6 (12.1 to 19.1) | -7.1% |
| KDE—single | | | 71.8* (69.3 to 74.2) | 46.5% | 55* (50.4 to 59.6) | -8.8% | | | | | | | | |
| KDE—aggregate | | | 74.3* (73.2 to 75.5) | 51.6% | 54.9* (50.4 to 59.5) | -9.0% | | | | | | | | |
| **Ecological—administrative** | | | | | | | | | | | | | | |
| Facility catchment and provider category* | | | 49.1 (43.6 to 54.6) | 0.2% | 59.8 (55.1 to 64.5) | -0.8% | | | | | | | | |
| Subdistrict and provider category* | | | 49.4 (43.9 to 54.9) | 0.8% | 57.9 (53.4 to 62.4) | -4.0% | | | | | | | | |
| District—unweighted† | 4.7* (21.4 to 31.7) | -52.5% | | | | | 34.9* (29.0 to 40.8) | -6.2% | 39 (32.3 to 45.7) | -2.7% | 17.8* (14.6 to 21.0) | -22.3% | 21* (17.2 to 24.8) | 25.0% |
| District—unweighted† and provider category* | 11.0 (3.8 to 18.2) | 11.8% | | | | | 37 (30.4 to 43.6) | -0.5% | 39.7 (32.7 to 46.7) | -1.0% | 20.3* (15.8 to 24.8) | -11.4% | 17.4 (13.3 to 21.4) | 3.6% |
| District—weighted‡ | | | | | | | 37.9 (31.3 to 44.4) | 1.9% | 40.7 (33.7 to 47.7) | 1.5% | 19.7* (16.1 to 23.2) | -14.0% | 21.2* (17.4 to 25.0) | 26.2% |
| District—weighted‡ and provider category* | | | | | | | 38.8* (31.9 to 45.7) | 4.3% | 40.8 (33.6 to 48.0) | 1.7% | 21.1* (16.4 to 25.8) | -7.9% | 17.1 (13.1 to 21.2) | 1.8% |

*Ecological linking restricted to only providers within the category (type of outlet, managing authority, and facility level) reported by survey respondent.
†Simple average of provider quality scores applied, not accounting for differentials in patient volume.
‡Provider quality scores weighted by provider utilisation volume for relevant health area.
QoC, quality of care.

**Table 4** Summary of evidence related to methodological issues for linking analyses and related needs for future research

**Suitability of household and provider data for linking analyses**
► Need valid data on target population for the intervention, and suitable data on service contact/care-seeking
► Need provider data reflective of select aspects of QoC, standardised indices and clear interpretation of measures

| Issue | Evidence | Action |
|---|---|---|
| How valid are data on target population for interventions? | ► Symptom/diagnosis-based conditions may be biased.<br>► Rare conditions are not captured with sufficient sample. | Explore alternative methods for defining population in need (eg, biomarkers, Bayesian modelling of disease probability). |
| How valid are data on care-seeking? | ► Limited data suggest respondent able to identify type of provider but not type of health worker.<br>► Inconsistent and sometimes poorly defined provider categories. | ► Validate care-seeking in more settings/health areas.<br>► Align categories of care across data collection tools. |
| How are QoC data being collected and what are the limitations of these methods? | ► Mostly through health facility surveys.<br>► HMIS data not widely used—limited QoC data collected.<br>► Alternative methods (record review, provider or client report, etc) correlate poorly with provision of services/process quality. | ► Assess validity of existing QoC measurement methods.<br>► Assess availability/usability of HMIS data for EC estimation.<br>► Develop and test new methods for assessing provision of care and experience of care. |
| How are quality measures being constructed and what do we know about the performance of these indices? | ► Mostly SPA/SARA structural data, limited indicators on provision or experience of care, EmONC signal functions.<br>► Variable set of indicators used based on guidelines and standards.<br>► Many methods for combining indicators have been tried.<br>► Handful of studies comparing methods produced conflicting results. | Develop standardised and validated summary QoC measures. |
| How well do measures of quality track with each other, clinical quality and/or health benefit? | ► Limited evidence of weak or no association between (1) structural and process quality, (2) measured quality and clinical care/health outcomes. | Standardise methods and terminology for defining and interpreting QoC measures to more accurately reflect role in the coverage cascade. |

**Implications of design of existing household and health provider data sources commonly used in linking analyses**
► DHS/MICS household location unknown, cluster location displaced and may introduce imprecision into ecological linking analyses.
► SPA/SARA often use sample of facilities and subsample of client–staff interactions that may not be representative of true service environment.
► Household and provider surveys are sampled and conducted independently → data are typically temporally and geographically discordant.

| Issue | Evidence | Action |
|---|---|---|
| Does imprecise DHS/MICS household location data affect ecological linking results? | Handful of studies suggest minimal effect on results produced by linking on geographical proximity. | Assess impact of household vs cluster centroid location vs displaced centroid in ecological linking analyses in multiple settings. |
| How does SPA/SARA sampling design affect estimates? | ► Two studies suggest impact of excluding non-facility providers is context specific.<br>► Client-staff interactions sampled to be representative at same level as overall survey—not at facility level.<br>► One study showed sampling of facilities resulted in moderate misclassification of service environment across linking methods.<br>► Joint sampling method proposed in 2001—oversample providers around sampled household clusters. | ► Assess effect of provider sampling (vs census) on linked estimates.<br>► Assess effect of within-facility sampling of healthworkers and client-healthworker observations.<br>► Triangulate with other sources of facility data (eg, HMIS) to take advantage of the greater detail of the SPA assessment with the bigger sample of the facility records.<br>► Account for uncertainty in estimates based on the facility-level data (eg, multilevel structure).<br>► Test alternative sampling methods to improve representativeness of provider survey sampling for clients and healthworkers.<br>► Test joint sampling methods for EC estimation. |
| How stable are indicators over time? | ► Studies demonstrate moderate indicator variability over months/years.<br>► No studies directly related to effect on linking analyses. | ► Assess stability of key provider and household indicators.<br>► Develop and test methods to account for unstable estimates, including more frequent data collection methods (eg, through HMIS) if needed. |

Continued

**Table 4** Continued

**Impact of choice of method for combining household and provider data**
► Multiple approaches for combining data sets, each with strengths and limitations.
► Exact match linking based on specific source of care most precise but ecological linking based on geographical proximity or administrative unit is more feasible.

| Issue | Evidence | Action |
|---|---|---|
| How do exact match and ecological linking approaches compare? | ► Three studies found ecological methods produced estimates similar to exact match under certain conditions in settings with high use of public providers.<br>► Restricting analyses by source of care category and/or weight by utilisation volume improved agreement with exact match. | ► Assess performance of ecological methods in settings with greater variation in provider landscape, provider quality.<br>► Define guidance, such as provider quality variation thresholds, for selection of linking method. |
| How do different ecological linking methods and measures of geographical proximity perform? | ► Similar results using straight-line, road distance and travel time.<br>► Variable performance of ecological methods in identifying true source of care/ reported category of care. | ► Identify preferred measures of geographical proximity to use in linking analyses.<br>► Create standard, accessible tools for conducting ecological linking. |
| What are the statistical challenges in combining data for effective coverage estimation? | ► Most analyses derive estimate variance from household sampling error.<br>► Two papers used delta method, but no comparison to other methods.<br>► Simulation found variance estimation using delta method performed better than household error alone or parametric bootstrapping. | Continue developing tools and approaches for estimating uncertainty around linked estimates. |

DHS, Demographic and Health Survey; EC, effective coverage; EmONC, emergency obstetric and newborn care; HMIS, Health Management Information Systems; MICS, Multiple Indicator Cluster Survey; QoC, quality of care; SARA, Service Availability and Readiness Assessment; SPA, Service Provision Assessment.

not individual healthcare worker cadre, but additional validation in various settings and service areas, such as postnatal care, would be informative. Further, it is essential to ensure that categorisation of sources of care in household surveys align with the categories used in provider assessments to facilitate linking datasets. The validity of household survey data for estimating populations in need was more variable. While some populations in need can be clearly defined, others, particularly those requiring symptom-derived diagnoses based on respondent report, have demonstrated potential for bias. Additional work is needed to explore alternative methods for identifying populations in need of interventions within population-based data sources.

The content and construction of provider quality indices was highly variable across publications, but largely derived from facility surveys and informed by international guidelines or recommendations. Methods for collecting provider quality have a number of limitations, and no single method perfectly encompasses all aspects of care.[80] The review found a lack of agreement between measures of quality derived through various means of collection. Overall, there was little empirical data supporting association between structural quality and process quality, and measures of quality and appropriate care or good health outcomes, although the number of reviewed studies was very limited. However, as articulated by Nguhiu *et al*, there is need to consider quality indicators' 'intrinsic value as levers for management action' and application to policy decision making in addition to their ability to capture or predict associated health gain.[38] Many important indicators of healthcare quality, particularly around patient-centred care, are not currently measured through existing tools and

there is a need to better capture these indicators.[81 82] Additional research is needed in the short term to develop and evaluate new quality indices using existing data sources (eg, facility surveys, HMIS and medical records) with an aim of identifying a standardised approach for selecting, combining, and interpreting indicators that reflect aspects of provider quality necessary for delivering appropriate, respectful and effective care. Longer term, substantial effort is needed to strengthen or adapt existing mechanisms or develop alternative methods for collecting provider quality indicators that can produce timely and informative estimates for tracking effective coverage of key interventions.

### Implications of the design of existing household and provider data sources commonly used in linking analyses
Few studies addressed the influence of the design of common data sources on linking analyses, including the impact of imprecise household GIS data, provider sampling frame and sampling design and estimate stability. However, there was a lack of concrete evidence around the impact of these factors on linked effective coverage estimates. Explicitly evaluating the impact of imprecise household location, sampling design and temporal gaps between measures within the context of effective coverage estimation would be informative. Mixed results on the inclusion of non-facility providers in provider assessments for effective coverage estimation emphasise the need to empirically assess the utilisation and service quality of non-facility providers in a given setting prior to conducting a linking analysis, as the quality and use of these providers varies by health area and setting.[70 71 83]

Although data related to impact on effective coverage estimation were limited, small samples of client-staff observations, sampling of health workers and facilities, and temporal gaps between household and provider data have the potential to bias estimates. The available data suggest that developing and testing alternative means of sampling health providers could improve the validity of linked estimates of effective coverage, including evaluating joint sampling approaches proposed by Measure Evaluation[67] or used by other data collection mechanisms such as Performance Monitoring for Action (PMA) and the India District Level Household and Facility Survey.

### Impact of choice of method for combining household and provider data

The most consistent evidence found through the review was around methods for combining data sets. Three papers compared ecological and exact match linking and reported that ecological linking (when accounting for frequency of provider utilisation by type) produced similar estimates to exact match linking. The agreement between the three publications that compared exact match and ecological linking is promising. Exact match linking is considered the most precise method for generating linked estimates; however, ecological linking is often more feasible because it does not require information on exact source of care or data on all providers. The papers further point to the need to maintain data on type of provider from which care was sought or the relative volume of patients seen by providers in order to generate valid estimates of effective coverage. All three studies were conducted in rural sub-Saharan Africa in settings with high utilisation of public sector health facilities; additional studies evaluating the performance of these methods in settings with a more diverse healthcare landscape would be informative. Other papers evaluated ecological linking approaches and found similar estimates of access to care or effective coverage using different approaches for assessing geographical proximity, although the ability of methods to capture true source of care was more variable. External to this review, additional data suggest that individuals may not always use the closest source of care and may bypass providers in favour of providers offering better care.[37 84 85] These findings along with the analyses comparing exact match and ecological linking approaches emphasise the need to carefully select methods for performing ecological linking and to control for true care-seeking behaviour as much as possible by accounting for the type of provider from which care was sought or weighting by utilisation in linking analyses. There is also need to further develop approaches and tools for estimating uncertainty around linked effective coverage estimates.

Evidence across the review demonstrates the need for careful choice of methods, data sources and indicators when conducting studies or analyses to link household and provider data for effective coverage estimation. An exploration of the precise effect of setting characteristics, such as variation in provider quality, on effective coverage estimates is needed to guide decision making in the selection of linking methods. Once more of these issues have been evaluated, additional tools and guidance to facilitate use of these methods will be needed.

The review was limited by the diversity of terminology and fields related to the linking methodology. However, the use of multiple search strategies minimised the likelihood of overlooking relevant publications. No formal grading of publication quality was included in the assessment, but the choice to conduct the search through Medline was intended to ensure a basic level of quality across the diverse study designs included in the review. Additionally, the diversity of fields, approaches and questions made it difficult to summarise the findings neatly, emphasising the need for communication between researchers, more standard terminology, and, ideally, a cohesive research strategy going forward. Recent efforts have aimed to align definitions of effective coverage.[2] We attempt in table 4 to translate the review results into actionable items and needs for future research.

### CONCLUSIONS

Linking household and healthcare provider data is a promising approach that leverages existing data sources to generate more informative estimates of intervention coverage and care. These methods can potentially address limitations of both household and provider surveys to generate population-based estimates that reflect not only use of services, but also the content and quality of care received and the potential for health benefit. However, there is need for additional research to develop evidence based, standardised best practices for these analyses. The most pressing priorities identified in this review are: (1) for those collecting data from health systems to explore methods to strengthen existing provider data collection mechanisms and promote temporal and geographical alignment with population-based measures, (2) for those collecting population-based data to address validity of self-reported intervention need and ensure indicators of access and utilisation of care are measured to facilitate linking analyses and (3) for those conducting linked analyses to standardise approaches for generating and interpreting effective coverage indicators, including sources of uncertainty, to ensure we are producing evidence that is harmonised, informative and actionable for governments and valid for monitoring population health globally.

**Contributors** ICMJE criteria for authorship read and met: EC, HL, TM, AA and MKM. Conceived of the study design: EC, MKM. Conducted review: EC. Drafted paper: EC. Agree with manuscript and conclusions: EC, HL, TM, AA and MKM. All authors read, edited and approved the manuscript.

**Funding** This work was supported, in whole, by the Improving Measurement and Program Design (IMPROVE) grant (OPP1172551) from the Bill & Melinda Gates Foundation. The funders did not have any role in the design of the study and collection, analysis, and interpretation of data or in writing the manuscript.

**Competing interests** None declared.

**Patient consent for publication** Not required.

**Provenance and peer review** Not commissioned; externally peer reviewed.

**Data availability statement** Data sharing not applicable as no datasets generated and/or analysed for this study. As a review article, this article reports data from previously published studies.

**ORCID iDs**
Emily D Carter http://orcid.org/0000-0003-1649-5274
Hannah H Leslie http://orcid.org/0000-0002-7464-3645
Melinda K Munos http://orcid.org/0000-0002-1349-8026

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
