## [Reviewer comments · BMJ Open]

ARTICLE DETAILS

TITLE (PROVISIONAL)	Methodological Considerations for Linking Household and Healthcare Provider Data for Estimating Effective Coverage: A Systematic Review
AUTHORS	Carter, Emily; Leslie, Hannah; Marchant, Tanya; Amouzou, Agbessi; Munos, Melinda

VERSION 1 – REVIEW

REVIEWER	Doshmangir, Leila Tabriz University of Medical Sciences
REVIEW RETURNED	04-Feb-2021

GENERAL COMMENTS	This is a valuable paper explores systematically methodological considerations for linking household and healthcare provider data for estimating effective coverage. The paper needs some revisions and clarifications. - The limitation part needs to be revised for example” Results of the review are summarized and related to actionable items and needs for future Research” .- The authors can use the sentence” Linking household and provider data could generate more informative estimates of effective coverage” as one of the recommendations or strengths of the study- In the abstract part I recommend to revise to revise conclusion section to answer the questions such as Is evidence reported with good quality? What did you learn? Now, results just reflect what is known.- The aim and objectives of the paper needs to more clarified. For example sentences like this “ linking household data with health care provider assessments is proposed as a means of generating more informative estimates of effective coverage, but methodological issues need to be addressed” .- Use the complete form of the words for first time use for example DHS and MICS or SPA and SARA.- The terms used for search are not consistent and related to the study subject for example benchmarking, system dynamics. I am not sure use of these terms could be useful in response to the study questions.- The results’ section related to the final included studies needs to be revised for example” 48 publications addressed a methodological concern...” / “ In total 62 publications addressed a methodological concern, including the suitability of household.... “. It is better to report final number of papers included to the study that address methodological concern
--

	 - What is the Data Analytics Approach in this systematic review? Did the authors quality appraisal? How did it? I recommend reporting results of quality appraisal. - How the themes/heads reported in the result section have been extracted or identified. Have the authors used content analysis approach? - The sentence “This review found a limited number of publications that explicitly addressed methodological issues I cannot understand due to the total number of papers(62 studies), why the authors have mentioned the number was limited. - Some findings were reported in the discussion section for example “Three papers compared ecological and exact match linking and reported that ecological linking.....” The discussion sections needs to be revised based on the results. Please just discuss the finding and not report the findings of the study in the discussion section. - I recommend adding key messages for health policy makers and planners.
--	---

REVIEWER	Cavallaro, Francesca University College London, Institute of Child Health
REVIEW RETURNED	23-Feb-2021

GENERAL COMMENTS	Thank you for the opportunity to review this systematic review of methodological considerations in linking population-based and provider data. The review was thorough and presents a coherent summary of very diverse methodological considerations, and is an important and timely contribution to this developing field. I have only a few minor comments below.  - The authors mention a previous systematic review by Do et al which seems closely aligned to the current systematic review. It might be helpful if the authors indicated in the introduction how the current review differs (in scope or time period considered) from previous ones - Results, paragraph 1: I am curious to know how many of the papers included in the reviews by Amouzou et al. and Do et al. were captured using the Medline/CPC/PHM/DHS searches? If most were, that would offer reassurance that the search strategy was well suited to the objectives, but if most were not it might suggest the use of longer phrases in the search strategy led to missing a high proportion of relevant publications. Having now looked at Figure 1, it seems at least 39 publications from DHS/Do/Amouzou were not captured in database searches, or 27 from the two reviews assuming the DHS publications were not indexed. Were these publications key ones, and were they retained for inclusion in the review? - “household surveys can be used to estimate the population in need of healthcare” p5: it would be helpful if the authors could make the distinction between absolute numbers vs. proportion of population in need here (or clarify that the population in need is within the population-based sample, since DHS/MICS to my knowledge do not provide cluster-level estimates of total population)
--

- Is the focus of the review on LMICs or all countries? It would be useful to clarify in the methods. (I had assumed the former, but then was surprised to see the USA mentioned in the results)

- Issues related to provider sampling “For provider assessments conducting direct observations of clinical care, the number and type of interactions observed within each health facility is dependent on patient volume and chance” – Perhaps not discussed in the papers reviewed, but I was surprised that the distinction was not made more clearly between sampling of providers for the provider surveys (of knowledge, training etc.) and the sampling of consultations for observation. Are the two sampling procedures independent for the DHS/SARA? From memory, it seems DHS reports do not include detailed descriptions of how they sample consultations and how they calculate the weights for these. I think it would strengthen the review to add a few sentences on this. And perhaps also that providers are sampled among those present at the facility on the day of the survey, which might bias estimates if, for example, (specialist) doctors hold consultations on specific days of the week.

- Table 4 is a useful summary and the actions recommended are in line with the findings.

- Discussion “Mixed results on the inclusion of non-facility providers in provider assessments for effective coverage estimation emphasize the need to empirically assess the utilization and service quality of non-facility providers in a given setting prior to conducting a linking analysis.” – if useful, the following reference shows that chemists are a popular source of care in several countries for family planning and childhood illnesses. Chakraborty, N.M., Sprockett, A. Use of family planning and child health services in the private sector: an equity analysis of 12 DHS surveys. *Int J Equity Health* 17, 50 (2018). <https://doi.org/10.1186/s12939-018-0763-7>

- Discussion “ecological linking produced similar estimates to exact-match linking under certain conditions” – it would be helpful to state what those conditions are if known. Also when mentioning bypassing of nearest provider, are there certain types of care that people are more likely bypass (e.g. delivery care) than others (e.g. sick child)? This might help inform which types of care-seeking behaviours are appropriate for ecological linking

VERSION 1 – AUTHOR RESPONSE

Reviewer: 1

Dr. Leila Doshmangir, Tabriz University of Medical Sciences

The limitation part needs to be revised for example” Results of the review are summarized and related to actionable items and needs for future Research” .

Added to the last paragraph of the discussion section “Recent efforts have aimed to align definitions of effective coverage [2]. We attempt in Table 4 to translate the review results into actionable items and need for future research.”

- The authors can use the sentence "Linking household and provider data could generate more informative estimates of effective coverage" as one of the recommendations or strengths of the study

The first sentence of the conclusions section begins with "Linking household and healthcare provider data is a promising approach that leverages existing data sources to generate more informative estimates of intervention coverage and care"

- In the abstract part I recommend to revise conclusion section to answer the questions such as Is evidence reported with good quality? What did you learn? Now, results just reflect what is known.

Add to conclusions section of the abstract: "However, existing evidence on methods for linking data for effective coverage estimation are variable and numerous methodological questions remain."

- The aim and objectives of the paper needs to more clarified. For example sentences like this "linking household data with health care provider assessments is proposed as a means of generating more informative estimates of effective coverage, but methodological issues need to be addressed".

The final sentence of the background section and first paragraph of the methods section present the objective of the review: "We conducted a systematic review to understand the current evidence base for effective coverage linking methods and identify needs for further research.

We searched for papers addressing methods or assumptions regarding: 1) the suitability of household and provider data used in linking analyses, 2) the implications of the design of existing household (DHS and MICS) and provider (SPA and SARA) data sources commonly used in linking analyses, and 3) the impact of choice of method for combining datasets to obtain linked coverage estimates."

- Use the complete form of the words for first time use for example DHS and MICS or SPA and SARA.

Corrected

- The terms used for search are not consistent and related to the study subject for example benchmarking, system dynamics. I am not sure use of these terms could be useful in response to the study questions.

The concept of "effective coverage" has evolved from related gap analyses. Terms such "systems dynamics" and "benchmarking" have been used in analyses similar to effective coverage cascades intended to understand barriers to improved health or gauge health system performance. These analyses are likely to utilize multiple data sources and potentially linking methods relevant to our review.

- The results' section related to the final included studies needs to be revised > for example "48 publications addressed a methodological concern..." / "In total 62 publications addressed a methodological concern, including the suitability of household....". It is better to report final number of papers included to the study that address methodological concern

The final number of papers included in the review, and counts by topic areas, are presented in the first paragraph of the results section. We have not repeated those numbers for the sake of reducing paper length.

- What is the Data Analytics Approach in this systematic review? Did the authors quality appraisal? How did it? I recommend reporting results of quality appraisal.

No formal quality analysis was conducted due to the varied designs and topics of papers included in the review. This has been clarified in Methods Paragraph 3: “No formal quality assessment was conducted due to the diversity of study designs and research objectives of the papers relevant to the review. Title and abstract review were conducted simultaneously by the first author (EC).”

- How the themes/heads reported in the result section have been extracted or identified. Have the authors used content analysis approach?

Added to last sentence of methods section: “Topical area groupings emerged from the review and were used to structure the findings.”

- The sentence “This review found a limited number of publications that explicitly addressed methodological issues I cannot understand due to the total number of papers(62 studies), why the authors have mentioned the number was limited.

Revised first sentence of discussion section to: “This review found a variable number of publications that addressed the diverse methodological issues related to linking household and provider datasets”

- Some findings were reported in the discussion section for example “Three papers compared ecological and exact match linking and reported that ecological linking.....” The discussion sections needs to be revised based on the results. Please just discuss the finding and not report the findings of the study in the discussion section.

The discussion section includes high-level summaries of key findings as needed to ground the discussion points.

- I recommend adding key messages for health policy makers and planners.

Our key message for health policy makers and planners are addressed in the conclusions section.

Reviewer: 2

Dr. Francesca Cavallaro, University College London

The authors mention a previous systematic review by Do et al which seems closely aligned to the current systematic review. It might be helpful if the authors indicated in the introduction how the current review differs (in scope or time period considered) from previous ones

Added to the second paragraph of the methods section: “Both the Do and Amouzou reviews summarized publications that linked data or estimated effective coverage; however, they did not systematically address methodological concerns or relevant results for guiding application of these methods.”

- Results, paragraph 1: I am curious to know how many of the papers included in the reviews by Amouzou et al. and Do et al. were captured using the Medline/CPC/PHM/DHS searches? If most were, that would offer reassurance that the search strategy was well suited to the objectives, but if most were not it might suggest the use of longer phrases in the search strategy led to missing a high proportion of relevant publications. Having now looked at Figure 1, it seems at least 39 publications from

DHS/Do/Amouzou were not captured in database searches, or 27 from the two reviews assuming the DHS publications were not indexed. Were these publications key ones, and were they retained for inclusion in the review?

Seven papers from the Amouzou and/or Do review were retained in our review. Only two of these papers were not captured in the Medline search (Marchant 2008 and Tanser 2006). Neither paper held essential information, the former addressed stability of commodity measures over time and the latter addressed the performance of travel time in approximating true source of care. Both were captured in our snowball approach as well.

- “household surveys can be used to estimate the population in need of healthcare” p5: it would be helpful if the authors could make the distinction between absolute numbers vs. proportion of population in need here (or clarify that the population in need is within the population-based sample, since DHS/MICS to my knowledge do not provide cluster-level estimates of total population)

Revised first sentence of first paragraph of “Suitability of household data needed for linked estimates” to say: “household surveys can be used to estimate the proportion of the population in need of healthcare”

- Is the focus of the review on LMICs or all countries? It would be useful to clarify in the methods. (I had assumed the former, but then was surprised to see the USA mentioned in the results)

Added to third paragraph of methods section: “The review focused on LMICs and data sources common in these settings, however publications from high-income settings were retained if the relevant evidence could translate to LMICs (e.g., use of centroid GPS location in estimates of distance, validity of provider quality measures).”

- Issues related to provider sampling “For provider assessments conducting direct observations of clinical care, the number and type of interactions observed within each health facility is dependent on patient volume and chance” – Perhaps not discussed in the papers reviewed, but I was surprised that the distinction was not made more clearly between sampling of providers for the provider surveys (of knowledge, training etc.) and the sampling of consultations for observation. Are the two sampling procedures independent for the DHS/SARA? From memory, it seems DHS reports do not include detailed descriptions of how they sample consultations and how they calculate the weights for these. I think it would strengthen the review to add a few sentences on this. And perhaps also that providers are sampled among those present at the facility on the day of the survey, which might bias estimates if, for example, (specialist) doctors hold consultations on specific days of the week.

Revised paragraph two of “Provider Sampling Design” section to: “A Measure Evaluation manual emphasized that data on provision of services (collected through observation of client-staff interactions), experience of care (collected through client exit interviews), and staff characteristics (collected through healthworker interviews) are sampled independently and collected among healthworkers and care interactions available on the day of the survey. They are a sub-sample of the overall survey and representative at the level the survey is sampled to be representative – not at the facility level”

- Table 4 is a useful summary and the actions recommended are in line with the findings.

Great!

- Discussion “Mixed results on the inclusion of non-facility providers in provider assessments for effective coverage estimation emphasize the need to empirically assess the utilization and service quality of non-facility providers in a given setting prior to conducting a linking analysis.” – if useful, the following reference shows that chemists are a popular source of care in several countries for family planning and childhood illnesses. Chakraborty, N.M., Sprockett, A. Use of family planning and child health services in the private sector: an equity analysis of 12 DHS surveys. *Int J Equity Health* 17, 50 (2018). <https://doi.org/10.1186/s12939-018-0763-7>

Added to paragraph 4 of the discussion section: “Mixed results on the inclusion of non-facility providers in provider assessments for effective coverage estimation emphasize the need to empirically assess the utilization and service quality of non-facility providers in a given setting prior to conducting a linking analysis, as the quality and use of these providers varies by health area and setting [72]”

- Discussion “ecological linking produced similar estimates to exact-match linking under certain conditions” – it would be helpful to state what those conditions are if known. Also when mentioning bypassing of nearest provider, are there certain types of care that people are more likely bypass (e.g. delivery care) than others (e.g. sick child)? This might help inform which types of care-seeking behaviours are appropriate for ecological linking

Changed second sentence of paragraph 5 of Discussion section to read: “Three papers compared ecological and exact match linking and reported that ecological linking (when accounting for frequency of provider utilization by type) produced similar estimates to exact-match linking”

VERSION 2 – REVIEW

REVIEWER	Cavallaro, Francesca University College London, Institute of Child Health
REVIEW RETURNED	04-May-2021

GENERAL COMMENTS	Thank you for the opportunity to review the revised version of this manuscript. It has been strengthened by extending the review time period, and the resulting nuancing of the conclusions. I cannot see a response to reviewers in the files available, but based on the marked copy I am satisfied that my minor concerns have been addressed and I have no further suggestions for improvement. This review will be a useful and timely tool for researchers in this area, and will hopefully contribute to defining methodological best practices for linkage of household and provider data.
--